# Resolved stress analysis, failure mode, and fault-controlled fluid conduits

David A. Ferrill[1], Kevin J. Smart[1], Alan P. Morris[1]

[1] Southwest Research Institute, 6220 Culebra Road, San Antonio, Texas 78238-5166, USA

*Correspondence to*: David A. Ferrill (dferrill@swri.org)

**Abstract.** Failure behaviours can strongly influence deformation-related changes in volume, which are critical in the formation of fault and fracture porosity and conduit development in low permeability rocks. This paper explores the failure modes and deformation behaviour of faults within the mechanically layered Eagle Ford Formation, an ultra-low permeability self-sourced oil and gas reservoir and aquitard exposed in natural outcrop in southwest Texas, U.S.A. Particular emphasis is placed on analysis of the relationship between slip versus opening along fault segments, and the associated variation in dilation tendency
versus slip tendency. Results show that the failure mode and deformation behaviour (dilation versus slip) relate in predictable ways to the mechanical stratigraphy, stress field, and specifically the dilation tendency and slip tendency. We conclude that dilation tendency versus slip tendency patterns on faults and other fractures can be analysed using detailed orientation or structural geometry data and stress information, and employed predictively to interpret deformation modes and infer volume change and fluid conduit versus barrier behaviour of structures.

**1      Introduction**

Faults and fractures often serve as conduits for fluid in low permeability rock (Barton et al., 1995; Caine et al., 1996; Zoback et al., 1996; Evans et al., 1997; Sibson and Scott, 1998; Ferrill and Morris, 2003; Faulkner et al., 2010; Alves and Elliott, 2014; Mattos et al., 2016; Mattos and Alves, 2018; Roelofse et al., 2020), including self-sourced oil and gas reservoirs (Ferrill et al., 2014a, 2014b, 2020; Gale et al., 2014), or $CO_2$ reservoirs (Trippetta et al., 2013; Ward et al., 2016; Miocic et al., 2020), and
reservoir cap-rock seals (e.g., Petrie et al., 2014; Roelofse et al., 2019). Permeability behaviour – flow pathway versus seal – can be directly related to the deformation modes along a fault, fracture, or fracture network (Carlsson and Olsson, 1979; Sibson, 1996, 1998, 2000, 2003; Trippetta et al., 2017; Ferrill et al., 2019a). In any applied stress field, multiple deformation features may form coevally, with failure initiation occurring at varying orientations and in different failure modes (e.g., Hancock, 1985; Lee et al., 1997; Lee and Wiltschko, 2000; Ferrill & Morris, 2003; Schöpfer et al., 2006; Busetti et al., 2014; Maher, 2014;
Smart et al., 2014; Douma et al., 2019; Boersma et al., 2020). Deformation behaviour, and in particular positive or negative dilation versus shear, is closely related to the orientation of the failure plane or zone with respect to the stress field at the time of deformation (e.g., Ramsey and Chester, 2004; Ferrill et al., 2017b). Recent work has shown that failure or reactivation mode along faults can be directly related to the dilation tendency versus slip tendency on the fault in the stress field at the time

of deformation (**Fig. 1**; Ferrill et al., 2012, 2017a, 2019b; Ward et al., 2016; Meng et al., 2020, Miocic et al., 2020; Roelofse et al., 2020).

In this paper, we explore the variability of resolved stress patterns along well-exposed and preserved, small displacement normal faults in the Eagle Ford Formation, and the relationship between dilation tendency, slip tendency, and deformation behaviour (failure mode) at various positions along faults following the approach presented by Ferrill et al. (2019a). The faults at the study site exhibit many segments that have measureable shear displacement and slickenlines, and other segments that have dilated and are partially or fully mineralized with calcite from the paleo-movement of aqueous fluids. Observations show that failure modes along individual faults can vary dramatically over distances of a few cm, governed by the lithologic changes and fault segment interaction. Shear versus dilational behaviour relates directly to the mechanical stratigraphy and the orientation of the failure zone within the stress field at the time of deformation (Ferrill and Morris, 2003). This study provides a clear example of how faults can serve as fluid conduits in mechanically layered low permeability strata. Furthermore, this work supports conclusions from seismic-scale observations that fault oversimplification misrepresents fault geometries and related damage zones, which translates to unreliable estimation of fault sealing behaviour (Ze and Alves, 2019). The use of dilation tendency versus slip tendency patterns shows significant potential for predicting failure or reactivation mode on faults or fractures, and the related conduit versus seal behaviour of those structures, applicable to detailed faults and fractures mapped or imaged in the subsurface.

## 1      Background

The Cretaceous Eagle Ford Formation has become an important self-sourced unconventional oil and gas reservoir in south Texas, U.S.A. (Robinson, 1997; Martin et al., 2011; Cusack et al., 2010; Bodziak et al., 2014; Breyer et al., 2016), and is an organic rich source rock for migrated oil produced out of other formations including the directly overlying Austin Chalk and underlying Buda Limestone (Edman and Pitman, 2010; Zumberge et al., 2016; Kornacki, 2018). In up-dip regions closer to the Eagle Ford outcrop belt along the Balcones fault zone, the Eagle Ford is an aquitard that forms a barrier to communication between aquifers including the overlying Austin Chalk, underlying Buda Limestone, and the deeper Edwards Aquifer (Livingston et al., 1936; Maclay and Small, 1983; Maclay, 1989; Ferrill et al., 2004, 2019b).

Analyses of the Eagle Ford oil and gas reservoir have shown the formation to have ultra-low permeabilities (50-1500 nanodarcies; Denney, 2012). This helps to explain the retention of self-sourced oil and gas in the formation, as well as the role

of the formation as a barrier to fluid movement. Recent outcrop studies, however, have shown that small-displacement faults – displacements of cm's to m's – within organic rich Eagle Ford Formation and overlying Austin Chalk that never reached oil window conditions necessary for hydrocarbon maturation are locally mineralized with calcite that contains fluorescent liquid hydrocarbon inclusions (Ferrill et al., 2014a, 2017a, 2020). Calcite cements in fault zone veins within the Eagle Ford Formation and Austin Chalk show crack-seal textures indicative of numerous incremental slip events, providing clear

indication of porosity generation and water movement from which the calcite precipitated (Ferrill et al., 2014a, 2017a, 2020). Migrated-oil inclusions in the calcite indicate longer distance up-dip travel of oil (likely tens of km) from areas where source rock strata reached oil generation conditions (Ferrill et al., 2020). Analyses of homogenization temperatures for two-phase (liquid-vapour) inclusions indicate fluid trapping at 1.4 to 2.9 km depths, and possibly as deep as 4.2 km (Ferrill et al., 2014a, 2017a, 2020). These trapping depth estimates indicate that the faults analysed here formed and remained active at these depths,

and are not near-surface phenomena. For comparison, these depths of normal fault formation and fluid movement are analogous to active fault controlled fluid flow based on 3D seismic interpretation in the Barents Sea (Mattos et al., 2016), North Sea (Alves and Elliot, 2014; Ward et al., 2016), and the Gulf of Mexico (Roelofse et a., 2020).

Refracted fault shapes and associated localization of dilation and cementation along these faults indicate the intricate interplay between mechanical stratigraphy and failure modes, and bed-to-bed switching in failure and reactivation behaviour

that led to formation of conduits for fluid flow through the otherwise very low permeability Eagle Ford Formation. Better understanding of the structural processes that influence formation of fault controlled fluid conduits is needed to evaluate migration and accumulation of hydrocarbons, as well as integrity of very low permeability sealing strata. Furthermore, this improved understanding could also aid interpretation of failure modes and fracture geometries produced by hydraulic fracturing in the Eagle Ford Formation and other mechanically layered unconventional reservoirs.


## 3    Methods

### 3.1    Fault segment characterization

Analyses in this paper focus on three faults in the Eagle Ford Formation exposed in bluffs along Sycamore Creek in southwest Texas. The three faults, in order of increasing displacement, are the (i) Textbook fault (max. throw in exposure = 10 cm; exposed height measured tip-to-tip = 7.2 m; **Fig. 2**), (ii) Spanish Goat fault (max. throw in exposure = 35 cm; exposed height from base of exposure to upper tip = 6 m), and (iii) Big Indigo fault (max. throw in exposure = 6 m; cuts entire 30 m height of exposure). These three faults are only exposed in the bluff, and cannot be mapped beyond the width of the cliff exposure. These faults were previously discussed and analysed by Ferrill et al. (2017a) and were selected from the larger population of faults at Sycamore Bluffs for detailed analysis because they (i) represent the spectrum of displacements on faults in the exposure, (ii) are in close proximity to each other, and (iii) represent faulting in the mudrock and chalk dominated pelagic reservoir section of the Eagle Ford Formation (Lehrmann et al., 2019). With respect to the measured section in Ferrill et al. (2017; their figure 3), measurements from the Textbook fault are from stratigraphic heights 1.25 m to 7.8 m, measurements from Spanish Goat fault are stratigraphic heights 4.6 m to 8.9 m, and measurements from the Big Indigo fault are from stratigraphic heights 5.35 m to 7.1 m. . Fault segments through different lithologic beds were mapped in the field directly onto digital photographs, and strike, dip, and rake (where slickenlines were visible) were measured using Brunton compass. Displacements were measured using metric measuring tape for the smaller displacement faults (i.e., Textbook fault, Spanish Goat fault, and NW segment of Big Indigo fault). The displacement of several meters and irregularity of the outcrop surface precluded direct field measurement of displacement on the main strand of the Big Indigo fault. Consequently, we surveyed the main trace of the Big Indigo fault using a spatial scanning system (Trimble VX™ Spatial Station) to measure the three-dimensional positions of offset marker beds at their hanging wall and footwall cutoffs, and from those data extracted displacements. The three faults analysed here exhibit significant changes in dip of the failure surfaces, and exhibit variation in deformation behaviour, including slickenlined slip surfaces and dilational segments that are partially or completely calcite-filled (**Fig. 2**).

## 3.2 Stress field interpretation

Stress inversion was performed using orientations of fault slip surfaces and displacement measurements from measured slip surfaces along the Textbook, Spanish Goat, and Big Indigo faults. The inversion was performed using the technique of McFarland et al. (2012) as implemented in 3DStress v. 5.1 (Morris et al., 2016). We adjusted the stress tensor solution slightly

to align the intermediate principal compressive stress ($\sigma_2$) orientation with the minimum eigenvector from the fault population because we expect $\sigma_2$ to be parallel to the intersection line direction for a conjugate normal fault population (Anderson, 1951; Thompson, 2015).

### 3.3 Dilation tendency and slip tendency analysis

Dilation tendency and slip tendency of a deformation feature or other fabric element are controlled by the orientations and relative magnitudes of the principal stresses in the imposed stress state. Dilation tendency ($T_d$) was defined by Ferrill et al. (1999) by the following Eq. (1):

$$T_d = \frac{(\sigma_1 - \sigma_n)}{(\sigma_1 - \sigma_3)}, \tag{1}$$

where $\sigma_n$ = resolved normal stress, $\sigma_1$ = maximum principal compressive stress, and $\sigma_3$ = minimum principal compressive stress. Slip tendency ($T_s$), was defined by Morris et al. (1996) by the following Eq. (2):

$$T_s = \frac{\tau}{\sigma_n}, \tag{2}$$

where $\tau$ = resolved shear stress. Dilation tendency and slip tendency analyses were performed in 3DStress, using the derived stress tensor and the measured orientations of the deformation features.

### 4 Results

### 4.1 Fault segment characterization

Data were collected representing 142 measurement positions along the three faults, tracking refracted faults through multiple lithologic layers. Orientation measurements were made from matching failure surfaces along both the hanging wall and footwall cutoffs for each measured bed cut by the fault (Textbook fault, number of beds measured (n) = 28; Spanish Goat fault, n = 23; Big Indigo fault, n = 20). The three faults represent progressive stages of increasing fault displacement and fault zone development. The faults have refracted fault profiles with numerous dip changes (e.g., **Fig. 2a**). These refracted profiles are represented not only by changes in dip, but also by changes in deformation behaviour, ranging from thin slip surfaces exhibiting slickenlines to thick calcite veins with crack-seal textures representing significant dilation and numerous dilational slip events. Some dilational fault segments are only partially filled with calcite cement and exhibit euhedral crystal terminations indicative

of crystal growth into open voids (**Fig. 2b**). Most of the dilational fault segments, however, are completely filled with calcite

(e.g., **Fig. 2c**), as described in Ferrill et al. (2017a). Calcite-cemented fault segments tend to have steep to vertical dips ($\geq75°$). In contrast, gently to moderately dipping fault segments ($<30\text{-}75°$) typically are marked with slickenlines, lack calcite cement, and reflect little or no positive dilation suggestive of shear or compactive-shear deformation. Measurement spacing (as a function of height of the survey portion of the fault) represents (i) 3.6% of the analyzed profile of the Textbook fault, (ii) 4.3% of the analyzed portion of the Spanish Goat fault (partial height of the fault), and (iii) 9.1 % of the analysed height of the Big

Indigo fault main trace (partial height of fault). A recent study by Ze and Alves (2019) mapped faults using 3D seismic reflection data and explored throw versus distance and throw versus depth profiles, and associated slip tendency and leakage factor analyses. Among Ze and Alves' (2019) conclusions were recommendations that sampling be performed at spacing of $<5\%$ for faults $< 3500$ m long, and $<3\%$ for faults $>3500$ m – our results would generally support these recommendations.

**4.2      Stress field interpretation**

The interpreted stress tensor used in the dilation tendency and slip tendency analyses is defined by the following relative magnitudes and orientations of the principal stresses as follows (**Fig. 3**): $\sigma_1 = 1.00$, azimuth 170°, plunge 71°; $\sigma_2 = 0.65$, azimuth 019°, plunge 17°; $\sigma_3 = 0.30$, azimuth 286°, plunge 9°. As noted earlier, we slightly adjusted the orientation of $\sigma_2$ from the initial stress inversion to align with the minimum eigenvector from the fault slip surface population (i.e., orientation

changed from 017°/25° to 019°/17°). We assume approximately 2 km of overburden, consistent with depths estimated from fluid inclusion analysis of veins from these and nearby faults (Ferrill et al., 2014a, 2017a, 2020); an average density of 2500 kg/m$^3$, consistent with the sedimentary overburden in the region; and pore pressure conditions (0.018 MPa/m), consistent with observations of overpressure in the Eagle Ford reservoir under production in south Texas. The three principal effective stress ($\sigma'$) magnitudes (adjusted for pore fluid pressure; cf. Ward et al., 2016) at the time of failure in mudrock are estimated to be

(i) $\sigma'_1 = \sim15$ MPa, (ii) $\sigma'_2 = \sim10$ MPa; and (iii) $\sigma'_3 = \sim5$ MPa. As noted earlier, dilation tendency and slip tendency are controlled by the orientations and relative magnitudes of the principal stresses in the imposed stress state, therefore a robust analysis can be performed without precise knowledge of stress magnitudes.

### 4.3    Dilation tendency and slip tendency analysis

The three faults investigated here each show a diverse spectrum of dilation tendency and slip tendency associated with their orientation (primarily dip but also strike) changes (**Fig. 4**). Comparing the dilation tendency and slip tendency profiles shows segments that fall into 3 primary categories: (i) low dilation tendency and low slip tendency, (ii) moderately high dilation tendency and high slip tendency, and (iii) high dilation tendency and low slip tendency. Cross plotting dilation tendency versus slip tendency for measured fault orientations (strike and dip measurements) shows this diverse spectrum of resolved stress

characteristics (**Fig. 5**). Differentiating between observations of calcite vein cement versus slickenlines associated with these fault segments shows a clear pattern of calcite vein cement associated with fault segments that have high dilation tendency and low to moderately high slip tendency (**✕** symbol in **Fig. 5**). Slickenlines were observed on fault segments that have moderately high to low dilation tendency and high to moderately low slip tendency (**+** symbol in **Fig. 5**). Depth intervals for measurement locations on the analysed faults are as follows: (i) Textbook fault = 0.05 to 0.6 m, (ii) Spanish Goat fault = 0.05 m to 0.4 m,

(iii) Big Indigo fault = 0.05 to 0.45 m. Systematic sampling more coarsely than this would underrepresent the fault irregularity and refraction that produced the dilation and localized fluid flow along the faults.

The moderate to high slip tendency and high dilation tendency of the steepest segments (dips >75°, red points) is consistent with hybrid failure. High slip tendency and moderately high dilation tendency for moderate to steep fault segments (dips of 45°-75°, green and gold points) is consistent with shear failure, whereas the moderate to low slip tendency and low dilation

tendency of the most gently dipping segments (dips <45°, light blue and dark blue points) is more consistent with compactive shear failure, although no definitive evidence of compactive shear (e.g., slickolites) was observed in the field (see **Fig. 1** for comparison). The pattern of dilation tendency versus slip tendency on the fault segments matches well with the deformation processes represented by vein material indicative of dilation versus absence of vein material and presence of slickenlines indicative of sliding (**Fig. 5**).


### 5    Discussion

Fault refraction through the mechanically layered Eagle Ford lithostratigraphic section led to conduit development at dilational segments along faults (**Fig. 6a**). These conduits are structurally controlled and form along the fault/bedding intersection within

more competent (chalk) beds where steep fault segments experienced dilation (dilational jogs; Ferrill and Morris, 2003; Ferrill et al. 2014a). Conduits tend to parallel the intermediate principal stress direction, which is horizontal or nearly horizontal in normal and thrust faulting stress regimes (e.g., Ferrill et al., 2019a, 2020), and vertical in a strike slip regime (Giorgetti et al., 2016; Carlini et al., 2019).

For a cohesionless fault, slip tendency at the initiation of slip is expected to equal the coefficient of friction on the fault (e.g., Byerlee, 1978; Morris et al., 1996). A fault that has slip tendency equal to the coefficient of friction is considered "critically stressed" (Stock et al., 1985; Barton et al., 1995; Morris et al., 1996; Zoback et al., 1996). This study shows that the slip tendency was highly variable along the refracted faults at the time of their active slip. Slip tendencies (**Fig. 5**) ranged from high values (>0.6), consistent with coefficients of friction of 0.6 to 0.85 described by Byerlee (1978), to low or very low values (0.4 to <0.2) on gently dipping fault segments that would make sense for activity only for low coefficients of friction associated with weak rock (see coefficient of friction summary in Ferrill et al., 2017b). Different rock types inherently have different friction coefficients, so a mechanical multilayer like the Eagle Ford Formation that includes chalk, marl, mudrock, and volcanic ash should be expected to have variable slip tendencies required to overcome the variable friction coefficients through different mechanical layers (**Fig. 6b**). The different mechanical properties of mudrock and chalk lead to different responses to loading conditions and produce significantly different pre-failure responses in mudrock versus chalk, and therefore different effective stress conditions from one mechanical layer to the next through the section. We are not specifically interpreting whether mudrock or chalk failed first. However, the repeated occurrence of refracted fault propagation through the section, contrasting mechanical properties of chalk and mudrock, and absence of widespread hybrid failure in chalk beds or shear failure in mudrock that is unassociated with larger multi-bed faults, suggests distinctly different effective stress conditions in mudrock and chalk shown in Fig. 6b likely coexisted in adjacent beds during fault propagation.

The clear relationships displayed in the dilation tendency versus slip tendency pattern, the failure and reactivation modes, and the mechanical layering, demonstrates the importance of understanding this interplay when investigating fault-related permeability development. Unconventional hydrocarbon reservoirs and low-permeability seal strata for aquifers, oil and gas reservoirs, and $CO_2$ reservoirs or sequestration sites are commonly not lithologically homogeneous, but instead are heterolithic and mechanically layered (e.g., Alves and Elliot, 2014; Petrie et al., 2014; Ward et al., 2016; Roelofse et al., 2019; Miocic et

al., 2020). Consequently, failure modes and failure orientations are likely to vary bed to bed and result in refracted fault shapes and fluid pathways similar to those discussed here.

The analysis in this paper clearly shows that deformation behaviour is intimately related to the orientation of the deformation feature with respect to the stress field in which it is active. Important orientation changes along the faults investigated here occur on the scale of individual beds over distances of cm's to 10's of cm. Generalized or smoothed fault shapes would not be representative of the actual behaviour of the fault. It is worth noting that fault refraction also occurs at much larger scales related to mechanical stratigraphy (see discussion in Ferrill et al., 2017b). To capture the important orientation variability that is critical to dilation tendency and slip tendency analysis requires careful mapping of the orientation changes in the maximum detail possible. Ze and Alves (2019) evaluated the influence of sampling on displacement characterization and segment identification for faults mapped with 3D seismic reflection data and concluded that a sampling interval on the scale of 3% to 5% of fault length was needed for robust analysis. As detailed mapping and close sample spacing is critical to identifying displacement changes and segments along faults (e.g., Wyrick et al., 2011; Ze and Alves, 2019), it is also critical to predicting the deformation behaviour using dilation tendency and slip tendency analysis.

The actual fault orientation variability described in the present study, however, is far too fine-scale to be mapped with seismic reflection data. Although the overall shape of a normal fault through the Eagle Ford Formation or similar rock that has throw of >10-20 m may be mappable from 3D seismic data, the bed-scale orientation variability along it will not be mappable. Using detailed mechanical stratigraphic characterization (e.g., from microrebound analysis of core), stress inversion, and understanding gained from this and other detailed investigations, failure mode prediction can help to bridge this gap and inform realistic representation of fault zone complexity.

## 6        Conclusions

Faults investigated here were active with refracted dip profiles and constituent segments that experienced widely varying dilation tendencies and slip tendencies at the time of activity. Deformation modes correlate with the dilation and slip tendency changes, and show that neither slip tendency nor dilation tendency alone are complete indicators of fault zone behaviour. The integrated analysis of dilation tendency and slip tendency, however, can be a very effective means to predict deformation

behaviour for fault segments or other structural features (fractures, layer boundaries, or mechanical interfaces). This

deformation behaviour is intimately related to the orientation of the deformation feature with respect to the stress field in which it is active, occurring on the scale of individual lithologic beds over distances of cm's to 10's of cm. To capture the important orientation variability that is critical to dilation tendency and slip tendency analysis requires careful mapping of the orientation changes in the maximum detail possible, and may require failure mode prediction based on detailed mechanical stratigraphic, stress, and geomechanical analysis informed by results of this and other detailed studies.


*Author contribution*: DAF was responsible for conceptualization, data curation, funding acquisition, methodology, and primary writing of the manuscript; KJS performed formal analysis and contributed funding acquisition and reviewing and editing of the manuscript; APM contributed to data collection for the investigation and methodology.

*Competing Interests.* The authors declare that they have no conflict of interest.

*Acknowledgements.* Slip and dilation tendency analyses were performed using 3DStress® v. 5.1. We thank Janice Moody and Heath Grigg for allowing us research access to the Rancho Rio Grande. Financial support for the field work was provided by Southwest Research Institute's Eagle Ford joint industry project, funded by Anadarko Petroleum Corporation, BHP Billiton,

Chesapeake Energy Corporation, ConocoPhillips, Eagle Ford TX LP, EP Energy, Hess Corporation, Marathon Oil Corporation, Murphy Exploration and Production Company, Newfield Exploration Company, Pioneer Natural Resources, and Shell. We thank the staff of these sponsor companies for the interaction and constructive feedback. Analyses in this manuscript were also supported in part by SwRI® Internal Research and Development Project R8940. We thank Ronald McGinnis, Dan Lehrmann, Erich DeZoeten, Sarah Wigginton, and Zach Sickmann for their contributions to data collection. Constructive

comments by Tiago Alves and Fabio Trippetta greatly improved the final version of this paper.

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

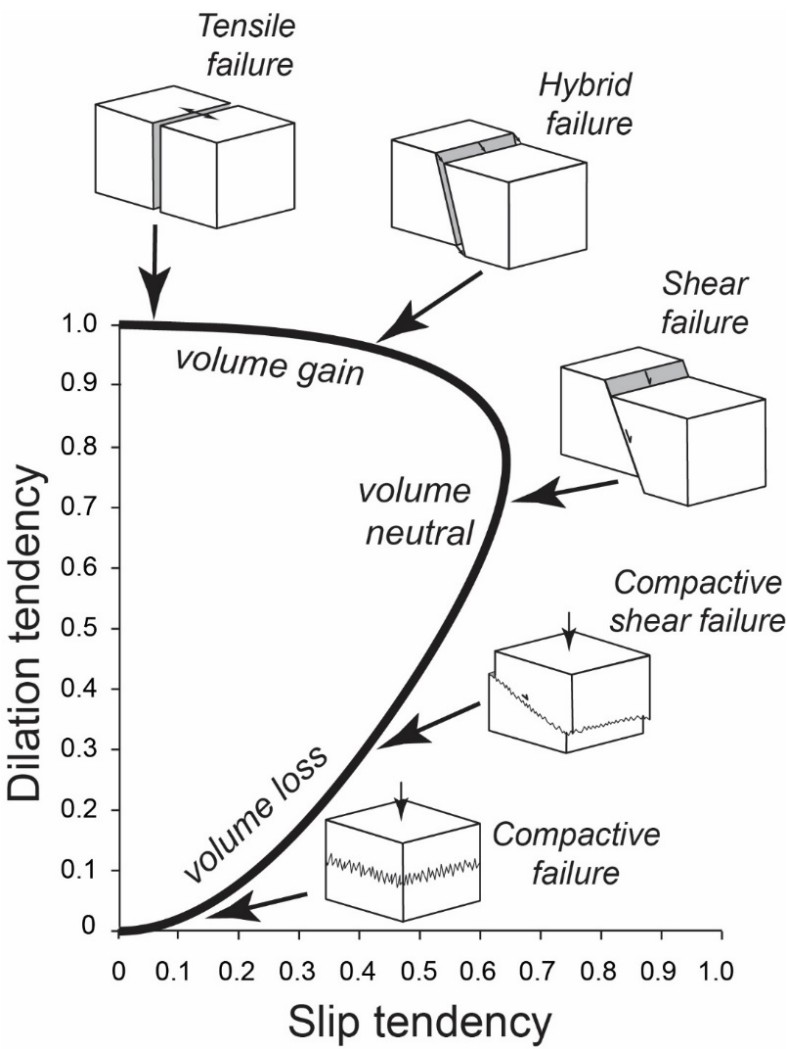

**Figure 1.** General graphical relationship between maximum slip tendency and dilation tendency, and associated rock failure

modes and volume change (from Ferrill et al., 2019a). As discussed by Ferrill et al. (2019a), analysing faults in this parameter

space shows promise for prediction of the failure or deformation modes and the associated conduit versus seal behaviour. For

purposes of this illustration, representative deformation features are shown for a normal faulting stress regime.

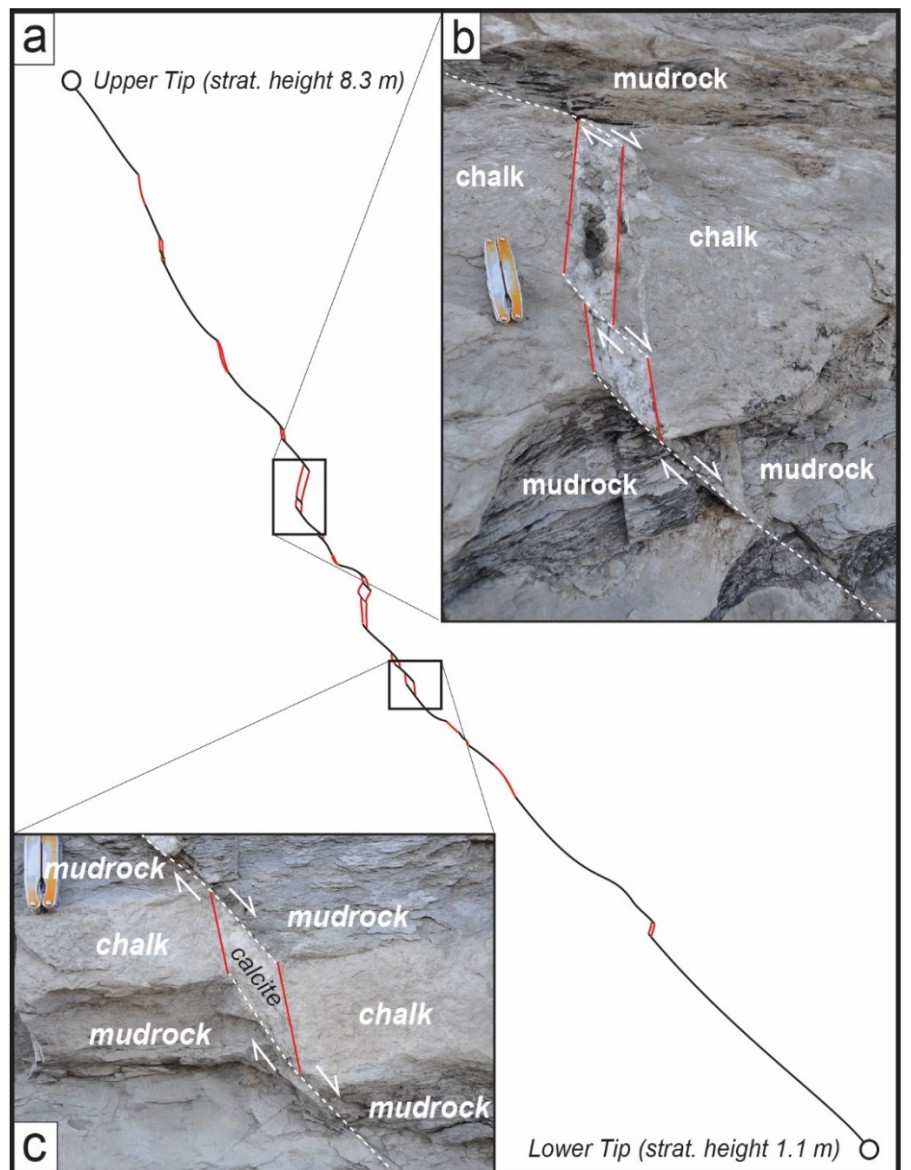

**Figure 2.** Details of the Textbook fault in Eagle Ford Formation outcrop at Sycamore Bluffs in southwest Texas. See Ferrill et al. (2017a) for additional detail on the exposure and faults. The section is heterolithic, including primarily chalk, marl, and calcareous mudrock. Faults tend to be represented by shear failure through mudrock, and hybrid failure through chalk beds, resulting in refracted fault profiles that exhibit dilation of steep segments through chalk beds. Dilational segments are represented by mechanical aperture that is partially cemented with calcite (see part **b**) or completely cemented with calcite (see part **c**).

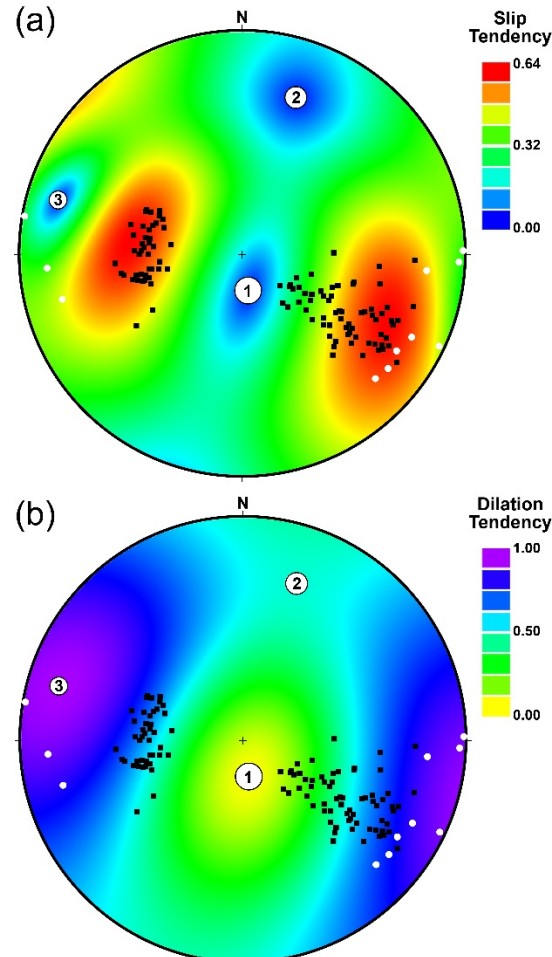

**Figure 3.** Equal-angle stereonet plots of (**a**) slip tendency and (**b**) dilation tendency (bottom) with poles to shear segments (black dots) and calcite cemented dilational segments (white dots) measured from the Textbook, Spanish Goat, and Big Indigo

faults at Sycamore Bluffs. Larger dots labelled 1, 2, and 3 represent orientations of the maximum, intermediate, and minimum principal compressive stresses, $\sigma_1$, $\sigma_2$, and $\sigma_3$, respectively. See text for further discussion of the inferred stress tensor.

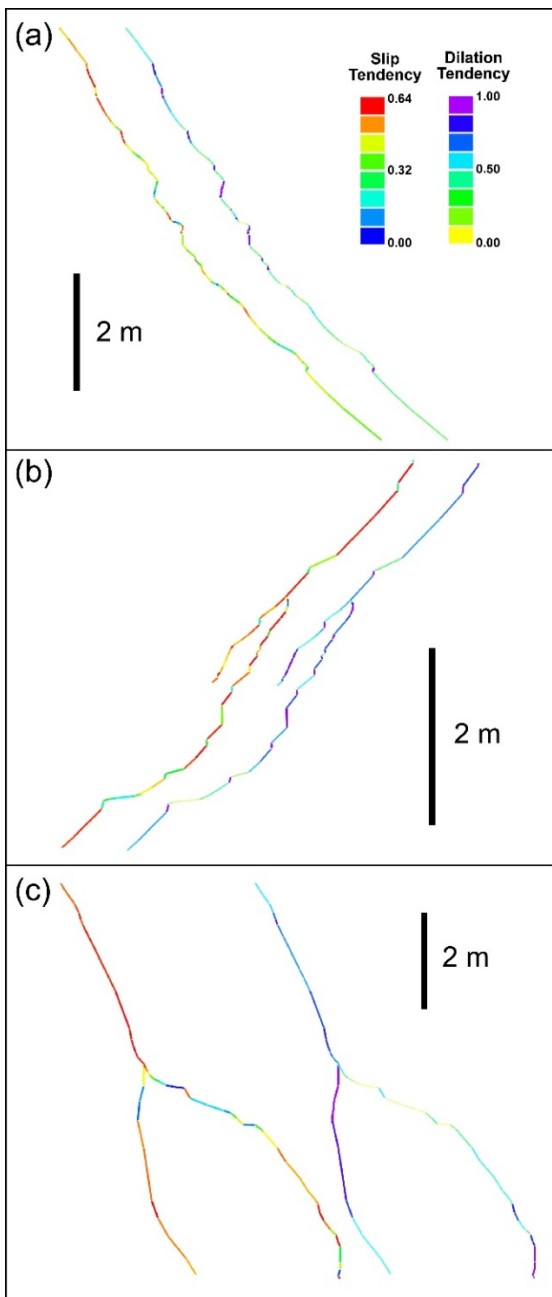

**Figure 4.** Slip tendency (left profile of each pair) and dilation tendency (right profile of each pair) profiles of the (**a**) Textbook,
(**b**) Spanish Goat, and (**c**) Big Indigo faults at Sycamore Bluffs using the inverted stress tensor described in the text and
illustrated in terms of slip tendency and dilation tendency in **Fig. 3**. Although these plots are similar to those presented in
Ferrill et al. (2017a), they have been updated to reflect the inverted stress state shown in **Fig. 3**.

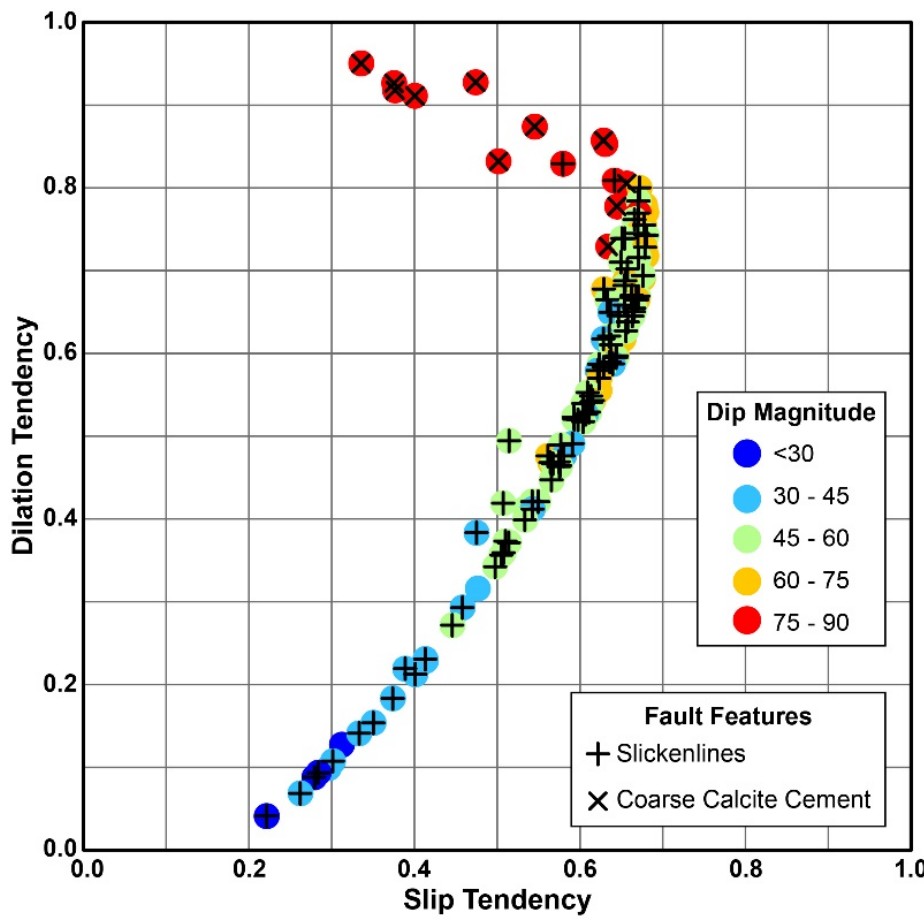

**Figure 5.** Comparison of dilation tendency and slip tendency for measured fault segments of the Textbook, Spanish Goat, and
Big Indigo faults, color-coded by dip, with + and × symbols indicating presence of slickenlines or coarse calcite cement,
respectively.  The few colored dots that lack additional symbols exhibit shear displacements, and either lack slickenlines or
slickenlines could not be seen due to the planar outcrop surface in some locations.  Moderate to high slip tendency and high
dilation tendency of the steep segments (dips >75°, red points) are consistent with hybrid failure, moderate to high slip tendency
and moderate dilation tendency of intermediate dips (45-75°, yellow and green points) are consistent with shear failure, and
the low to moderate slip tendency and low dilation tendency of the most gently dipping segments (dips <45°, light blue and
dark blue points) are suggestive of compactive shear.  See **Fig. 1** for comparison.

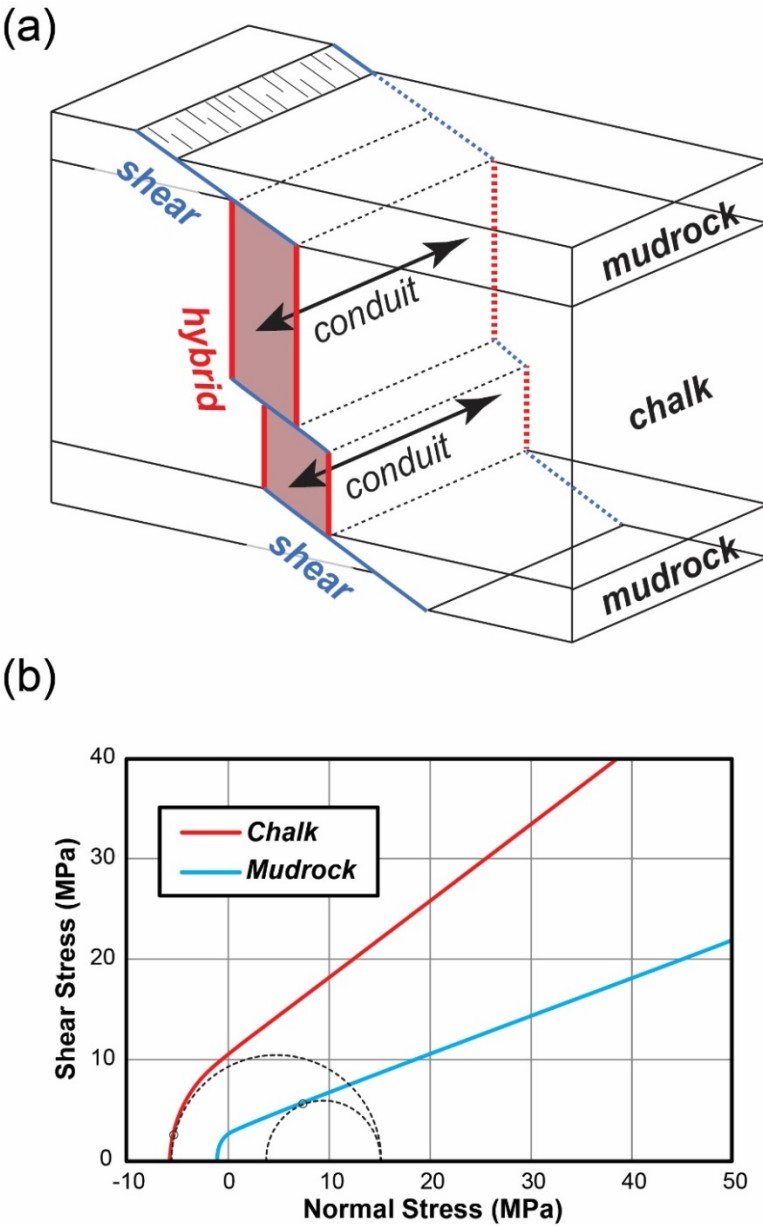

**Figure 6.** (**a**) Schematic block diagram illustrating the change in failure angle and mode from mudrock to chalk, and the associated dilation of the steeper hybrid segment and formation of a fault conduit parallel to the fault-bedding intersection direction. (**b**) Interpreted failure envelopes and stress circles for chalk and mudrock at the time of failure with a uniform effective overburden stress of 15 MPa (corrected for pore fluid pressure) with hybrid failure predicted for the more competent chalk beds and shear failure predicted for the less competent mudrock.