# Peer review of "Resolved stress analysis, failure mode, and fault-controlled fluid conduits"

_Solid Earth, 2020_

## Referee Comment (RC1) · Tiago Alves (Referee) · 19 Feb 2020

Dear authors

I liked reading your paper, particularly after acknowledging that the analysis in this work is similar to that we have developed at Cardiff since we first contacted SWRI in 2013 - and collaborated with this latter institute. With this in mind, se-2020-17 is an excellent addition to what has been an attempt at characterising fault-related fluid flow using high-quality seismic data. I was very pleased with having a field analogue of what we see on seismic.

[Figure]

I think this paper needs a moderate revision, and I appended an annotated .pdf to this review. The main points to be improved are:

Title - is the analysis in this work only valid for low-permeability data? I feel the analysis is broader than the title suggests.

1-Very old references are used at the start of the paper. Why such broader references when the paper is very much about fault slip and associated tendency to leak?

2 and 3-Seismic-based analyses have been undertaken by N. Ward et al. (2016). Tectonophysics and Roelofse et al. (2019) in basins posed for CO2 capture and storage. I would suggest the authors to indicate that low-permeability intervals have been characterised in detail using high-quality seismic data and borehole information.

4-Case studies are missing at the end of Page 1.

5 & 6 - This part hints at the problem of scale in fault segment interaction. At what scale this interaction occurs? Could you kindly complete this introduction with the comments and ideas in Tao and Alves (2017) Reply letter and Tao and Alves (2019). Tectonophysics? These are important papers that review the importance of understanding fault segment length at several scales of analysis - without under-interpreting data - as fluid flow will be controlled by elusive roughness, pull aparts and local refraction features in faults. It is reassuring to see this paper (se-2020-17) confirm the aspects in Tao and Alves (2017; 2019).

7-Once again, examples exist of similar approaches in Ward et al. (2016) and Roelofse et al. (2019). Mattos et al. (2016; 2018) are also interesting papers from Cardiff.

8-low permeability, rather than 'impermeable' strata. There is no such thing as impermeable strata.

9-Total/maximum lengths of faults need to be stressed at the start of the paragraph. Which stratigraphic section? Detail needed.

10 (page 8) - The 'scale problem' arises once again. Do the fault segments obey the rules in Tao and Alves (2019) that we need to collect T/Z data at a minimum spacing of 5% of a fault zone length to identify the presence of discrete segments; otherwise faults will resemble large constant-length structures? I am not asking for the inclusion of T/Z data in your paper, but it would be good to understand if the 5% rule is clearly recognised in the field - note: some longer faults require T/Z measurements at 3% of the length of a fault zone so that one can identify discrete segments. I think 2-3 paragraphs confirming how the segments are identified in se-2020-17 is very important in this page 8.

11-Add examples with work undertaken by the Cardiff group using seismic data. The stress tensors are rather similar to some of our work.

12-Segment scale needs to be referred to once again. Do they obey the field observations, which are seemingly based on the recognition of linkage points and inflexion/trend changes in discrete fault segments? (see Tao and Alves, 2017 Reply).

In essence, I overly enjoyed to read this work. The comments above will broaden the scope of this paper - particular those referring to the scale of fault and joint segments in the field and the way(s) they are recognised.

Please also note the supplement to this comment:
https://www.solid-earth-discuss.net/se-2020-17/se-2020-17-RC1-supplement.pdf

---

## Referee Comment (RC2) · Fabio Trippetta (Referee) · 19 Mar 2020

The paper "Resolved stress analysis, failure mode, and fault-controlled fluid conduits in low-permeability strata" by Ferril et al., is well organized and deals with the very interesting topic of mechanical models (generally speaking) where the authors are very expert. I really enjoyed reading it.

The paper follows some previous works extending theoretical models to real faults deeply studied previously by the same authors. In light of this, the boundary conditions of the applied mechanical model should be very well explained and constrained, in my opinion, in order to give to the reader all the instruments to completely understand the

meaning of the results. This is the part of the paper that I think should be improved.

In particular it is not clear to me for example what are the constrains for the hypothesized pore pressure, being this quite high (lambda over 0.7). The same for the mechanical properties of the involved lithologies proposed in Figure 6. No indication is reported along the paper about the source for the adopted mechanical data such as for example cohesion and coefficient of friction.

Keeping the focus on Fig.6 the proposed model is not clear to me. Since no bulid-up processes for fluid pressure are invoked along the text, if I well understand, rocks will fail in the initial stage, for a decrease of the sigma 3 being the system in an extensional regime. This bring mudrock to break first as showed in the model. Thus, at this time, a decrease in pore pressure is expected since, generally speaking, a rupture is related to an increase in permeability/porosity that lead to a decrease in pore pressure. However, following the model, a continuous process of build up for fluid pressure should be present in the system in order to overcome the sigma 3 and bring to hydraulic fractures on chalk. So I am wondering how can we reach the condition for high overpressure on chalk if a rupture already occurred on mudrock?.

That said, should we assume different boundary conditions for mudrock and chalks and reconsider figure 3?

In conclusion I think that this very interesting paper deserves some more rigorous constrains for the applied mechanical model. Moreover, a more comprehensive discussion on the model implication and on its evolution over time and space will strongly improve the paper together with a comparison with results form other authors (see line to line comments).

Some line to line notes are on the pdf attached file

Hope this helps

Fabio Trippetta

Please also note the supplement to this comment:
https://www.solid-earth-discuss.net/se-2020-17/se-2020-17-RC2-supplement.pdf

**Supplement:**

[revised manuscript text omitted]

---

## Author Comment (AC1) · 2 Apr 2020

Comment – "Dear authors, I liked reading your paper, particularly after acknowledging that the analysis in this work is similar to that we have developed at Cardiff since we first contacted SWRI in 2013 - and collaborated with this latter institute. With this in mind, se-2020-17 is an excellent addition to what has been an attempt at characterizing fault-related fluid flow using high-quality seismic data. I was very pleased with having a field analogue of what we see on seismic."

Author's Response – Thank you for the positive feedback!

[Figure]

Author's Change in Manuscript – Revised manuscript will enhance comparison to published examples and observations based on high-quality seismic data.

Comment – I think this paper needs a moderate revision, and I appended an annotated .pdf to this review.

Author's Response – Accept.

Author's Change in Manuscript – Manuscript revision will address comments in annotated .pdf provided by reviewer.

Comment – Title - is the analysis in this work only valid for low-permeability data? I feel the analysis is broader than the title suggests.

Author's Response – Accept, good point.

Author's Change in Manuscript – Title is being shortened by removal of "in low-permeability strata" as suggested by reviewer.

Comment – 1-Very old references are used at the start of the paper. Why such broader references when the paper is very much about fault slip and associated tendency to leak?

Author's Response – Accept.

Author's Change in Manuscript – Additional more recent relevant references are being added in the revision, as suggested by the reviewer.

Comment – 2 and 3-Seismic-based analyses have been undertaken by N. Ward et al. (2016). Tectonophysics and Roelofse et al. (2019) in basins posed for CO2 capture and storage. I would suggest the authors to indicate that low-permeability intervals have been characterized in detail using high-quality seismic data and borehole information.

Author's Response – Accept.

Author's Change in Manuscript – Additional relevant references for CO2 sequestration

and examples characterized using seismic and borehole information are being included in the revision, as suggested by the reviewer.

Comment – 4-Case studies are missing at the end of Page 1.

Author's Response – Accept.

Author's Change in Manuscript – Case studies from the literature are being added in support of this, as suggested by reviewer.

Comment – 5 & 6 - This part hints at the problem of scale in fault segment interaction. At what scale this interaction occurs? Could you kindly complete this introduction with the comments and ideas in Tao and Alves (2017) Reply letter and Tao and Alves (2019). Tectonophysics? These are important papers that review the importance of understanding fault segment length at several scales of analysis - without under-interpreting data - as fluid flow will be controlled by elusive roughness, pull aparts and local refraction features in faults. It is reassuring to see this paper (se-2020-17) confirm the aspects in Tao and Alves (2017; 2019).

Author's Response – Accept.

Author's Change in Manuscript – Revised manuscript will expand on this and include the relevant suggested references.

Comment – 7-Once again, examples exist of similar approaches in Ward et al. (2016) and Roelofse et al. (2019). Mattos et al. (2016; 2018) are also interesting papers from Cardiff.

Author's Response – Accept.

Author's Change in Manuscript – Revised manuscript will expand on this and include the relevant suggested references.

Comment – 8-low permeability, rather than 'impermeable' strata. There is no such thing as impermeable strata.

Author's Response – Accept.

Author's Change in Manuscript – Revised text will used adjusted language as suggested.

Comment – 9-Total/maximum lengths of faults need to be stressed at the start of the paragraph. Which stratigraphic section? Detail needed.

Author's Response – Accept.

Author's Change in Manuscript – Requested detail will be included in the revised manuscript.

Comment – 10 (page 8) - The 'scale problem' arises once again. Do the fault segments obey the rules in Tao and Alves (2019) that we need to collect T/Z data at a minimum spacing of 5% of a fault zone length to identify the presence of discrete segments; otherwise faults will resemble large constant-length structures? I am not asking for the inclusion of T/Z data in your paper, but it would be good to understand if the 5% rule is clearly recognized in the field - note: some longer faults require T/Z measurements at 3% of the length of a fault zone so that one can identify discrete segments. I think 2-3 paragraphs confirming how the segments are identified in se-2020-17 is very important in this page 8.

Author's Response – Accept.

Author's Change in Manuscript – The revised manuscript will discuss the identification of discrete segments as suggested.

Comment – 11-Add examples with work undertaken by the Cardiff group using seismic data. The stress tensors are rather similar to some of our work.

Author's Response – Accept.

Author's Change in Manuscript – Revised manuscript will make reference to this other published work as suggested and appropriate.

Comment – 12-Segment scale needs to be referred to once again. Do they obey the field observations, which are seemingly based on the recognition of linkage points and inflexion/trend changes in discrete fault segments? (see Tao and Alves, 2017 Reply).

Author's Response – Accept.

Author's Change in Manuscript – Segment scale will be addressed in the revised manuscript

Comment – In essence, I overly enjoyed to read this work. The comments above will broaden the scope of this paper - particular those referring to the scale of fault and joint segments in the field and the way(s) they are recognized.

Author's Response – Accept – thank you!

Author's Change in Manuscript – Revisions to the manuscript will broaden the scope as recommended.

Comment – Please also note the supplement to this comment: https://www.solid-earth-discuss.net/se-2020-17/se-2020-17-RC1-supplement.pdf

Author's Response – Accept.

Author's Change in Manuscript – The marked up manuscript supplement is being consulted in addressing the revie

---

## Author Comment (AC2) · 2 Apr 2020

Comment – The paper "Resolved stress analysis, failure mode, and fault-controlled fluid conduits in low-permeability strata" by Ferrill et al., is well organized and deals with the very interesting topic of mechanical models (generally speaking) where the authors are very expert. I really enjoyed reading it.

Author's Response – Accept – Thank you for the positive feedback!

Author's Change in Manuscript – No change needed to address this comment.

Comment – The paper follows some previous works extending theoretical models to

real faults deeply studied previously by the same authors. In light of this, the boundary conditions of the applied mechanical model should be very well explained and constrained, in my opinion, in order to give to the reader all the instruments to completely understand the meaning of the results. This is the part of the paper that I think should be improved.

Author's Response – Accept.

Author's Change in Manuscript – Additional detail to address stress and geomechanical assumptions and interpretations will be included in the revised manuscript.

Comment – In particular it is not clear to me for example what are the constraints for the hypothesized pore pressure, being this quite high (lambda over 0.7). The same for the mechanical properties of the involved lithologies proposed in Figure 6. No indication is reported along the paper about the source for the adopted mechanical data such as for example cohesion and coefficient of friction.

Author's Response – Accept.

Author's Change in Manuscript – Additional explanation of pore pressure, stress, and geomechanical assumptions and interpretations will be included in the revised manuscript.

Comment – Keeping the focus on Fig. 6 the proposed model is not clear to me. Since no build-up processes for fluid pressure are invoked along the text, if I well understand, rocks will fail in the initial stage, for a decrease of the sigma 3 being the system in an extensional regime. This bring mudrock to break first as showed in the model. Thus, at this time, a decrease in pore pressure is expected since, generally speaking, a rupture is related to an increase in permeability/porosity that lead to a decrease in pore pressure. However, following the model, a continuous process of build up for fluid pressure should be present in the system in order to overcome the sigma 3 and bring to hydraulic fractures on chalk. So I am wondering how can we reach the condition for

high overpressure on chalk if a rupture already occurred on mudrock. That said, should we assume different boundary conditions for mudrock and chalks and reconsider figure 3?

Author's Response – Because of different mechanical properties of mudrock and chalk, response to loading conditions produces significantly different pre-failure responses in mudrock versus chalk, and therefore different effective stress conditions from one mechanical layer to the next through the section. We are not specifically interpreting whether mudrock or chalk failed first. However, the repeated occurrence of refracted fault propagation through the section, contrasting mechanical properties of chalk and mudrock, and absence of widespread hybrid failure in chalk beds or shear failure in mudrock that is unassociated with larger multi-bed faults, suggests distinctly different effective stress conditions in mudrock and chalk shown in Fig. 6b likely coexisted in adjacent beds during fault propagation.

Author's Change in Manuscript – Text will be modified in the manuscript revision to further clarify this point.

Comment – In conclusion I think that this very interesting paper deserves some more rigorous constraints for the applied mechanical model. Moreover, a more comprehensive discussion on the model implication and on its evolution over time and space will strongly improve the paper together with a comparison with results from other authors (see line to line comments).

Author's Response – Accept.

Author's Change in Manuscript – Thank you for the positive feedback. Additional discussion and references will be included in the revised manuscript, as suggested by reviewer.

Comment – Some line to line notes are on the pdf attached file. Hope this helps, Fabio Trippetta Please also note the supplement to this comment: https://www.solid-earthdiscuss.net/se-2020-17/se-2020-17-RC2-supplement.pdf

Author's Response – Accept – thank you.

Author's Change in Manuscript – The marked-up manuscript supplement is being consulted in revising the manuscript.